# Improving routine immunization data quality using daily short message system reporting platform: An experience from Nasarawa state, Nigeria

Adekunle Akerele[1]*, Belinda Uba[1], Matthew Aduloju[1], Sulaiman Etamesor[2], Jamila A. Umar[2], Olorunsogo Bidemi Adeoye[1], Ameh Enyojo[1], Friday Josiah[1], Esther Ayandipo[1], Itse Olaoye[1], Oluwasegun Joel Adegoke[3], Sampson Sidney[4], Murtala Bagana[2], Okposen Bassey[2], Margherita E. Ghiselli[3], Waziri Ndadilnasiya[1], Omotayo Bolu[3], Faisal Shuaib[2]

1 African Field Epidemiology Network, Abuja, Nigeria, 2 National Primary Health Care Development Agency, Abuja, Nigeria, 3 Centers for Disease Control and Prevention, Atlanta, Georgia, United States of America, 4 Sydani Initiative for International Development, Abuja, Nigeria

* kunleakerele@gmail.com

**Data Availability Statement:** All relevant data are contained within the paper and its supporting information.

## Abstract

Routine immunization (RI) delivery was declared a public health concern in Nigeria in 2017 because of persistently low immunization coverage rates reported in independent surveys. However, administrative coverage rates remain high, suggesting serious data quality issues. We posit that a shorter timespan between service provision and data reporting can improve the monitoring of RI data, and developed a short message system (SMS) text reporting strategy to generate daily RI data points from health facilities (HFs). The goal was to assess whether daily data collection produces complete, reliable and internally consistent data points. The SMS reporting platform was piloted between December 2017 and April 2018 in two Local Government Areas (LGAs, equivalent to districts) of Nasarawa state, Nigeria. The 145 healthcare workers from 55 HFs received one mobile phone and pre-configured SIM card, and were trained to send data through predefined codes. Healthcare workers compiled the data after each vaccination session and transmitted them via SMS. We analyzed completeness, number of weekly sessions, and supportive supervision conducted. During the pilot phase, we received data from 85% (n = 47) of the 55 HFs. We expected 66 fixed-post sessions and 30 outreach sessions per week, but received data for 33 fixed-post and 8 outreach weekly session on average. More HFs reported on Tuesdays compared to other days of the week. When assessing internal consistency, we observed that the reported number of children vaccinated was sometimes higher than the number of doses available from opening a given number of vaccine vials. When found, this discrepancy was noted for all antigens during fixed-post and outreach vaccination sessions. Despite these initial discrepancies, transmitting RI data sessions via texting is feasible and can provide real-time updates to the performance of the RI services at the HF level.

**Funding:** This work was supported by the Center for Disease Control, Atlanta, in the form of a pool of funds to the African field epidemiology network to support the Nigerian government through the Nigerian National Primary Health Care Development Agency. Dedicated staff for project management was provided by the Nigerian National Primary Health Care Development Agency. No additional external funding was received for this study.

**Competing interests:** The authors have declared that no competing interests exist on this study.

## Introduction

Immunization against childhood diseases such as diphtheria, pertussis, tetanus, polio and measles are one of the most cost-effective health interventions, preventing childhood morbidity and mortality, especially among children younger than 5 years [1]. In most countries, routine immunization (RI) services are part of the primary health care system [2]. Increases in a country's immunization coverage often serve as an indicator of improvement of a health system's capacity to deliver essential services to the population [3]. In Nigeria, a child is considered fully vaccinated if he or she has received: 1 dose of Bacille Calmette-Guérin (BCG) vaccination against tuberculosis, 3 doses of Pentavalent (Penta) vaccine to prevent diphtheria, pertussis (whooping cough), tetanus, hepatitis B and *Haemophilus influenzae* type b, 3 doses of oral poliovirus vaccine (OPV), 1 dose of inactivated poliovirus vaccine (IPV), 1 dose of measles vaccine and 1 dose of yellow fever vaccine [1]. Children should receive these antigens during the first year of life.

The National Primary Health Care Development Agency (NPHCDA) of the Federal Ministry of Health (FMoH) of Nigeria coordinates RI policies and strategies while the Local Government Areas (LGAs, equivalent of districts) are responsible for the implementation of RI services [3]. Community access to RI services is promoted through the implementation of at least one fixed-post vaccination session per week, as well as at least one outreach session per week to offer immunization services to residents who live in remote areas. Despite these efforts, results from the 2016 Multiple Indicator Coverage Survey/National Immunization Coverage Survey (MICS/NICS) report that Penta 3 coverage at the national level was 33%, with variation between 3% and 80% across the 36 states and the Federal Capital Territory (FCT) [4]. This result is considerably different from the national administrative coverage rate of 78% reported for 2016. This survey result suggests that administrative RI data do not reflect true vaccination coverage for Nigeria. Several factors can contribute to inaccurate and inconsistent administrative data. Examples are erroneous data reporting from health facilities (HFs), incomplete data reporting, delays in recording summarizing vaccination data after RI sessions and poor review and use of RI data locally [5].

To address these challenges, NPHCDA established the National Emergency Routine Immunization Coordination Center (NERICC) in July 2017. NERICC is a body of governmental personnel and partners tasked with identifying the root causes of low RI coverage, developing strategies to address these challenges and coordinating with state- and LGA-level officials to implement these strategies. One of the proposed strategies was daily reporting of conducted RI sessions using a short message service (SMS) communication methodology.

In Nigeria, SMS communication methodology has been applied in many intervention, including reminder SMS sent to mothers in rural communities on full and timely completion of routine childhood immunization [4] and communication on specific health interventions for purposes of behavioral change [5]. In systematic review conducted on studies using SMS technology, researchers found out that it helped to enhance frequency of reporting health issues, improved patient medical compliance and appointment reminders, easy to use, relative inexpensiveness and automated message delivery [5–7]. However, there is no evidence of its use as a reporting platform for RI data from a HF level.

This paper describes the activities put in place to implement a pilot study and test the effectiveness of transmitting RI data via SMS texts. The assumption underlying this project is that data recorded daily at point of service will improve monitoring of data quality compared to data aggregated once per month over multiple RI sessions. Although immunization data are expected to be recorded on the day of administration at HFs, if they are not, error in recall increases as more time passes between the events of interest and their reporting [8–10]. Daily

surveys are often used to collect the most accurate and reliable prospective behavioral data, reduce the risk of selective reporting, and significantly reduce data management efforts [11–13]. Also, data are less likely to be missing entry when using real-time electronic data collection tools compared to paper-based tools [13]. Finally, recent studies found that participants prefer electronic data collection methods to paper-based strategies [13]. Therefore, we posit that daily real-time reporting of RI data through SMS can benefit program monitoring and overall RI data quality.

We described the launch the RI data collection system by SMS texting in Nasarawa state, Nigeria and report the first data points obtained through this new channel. This includes describing the components of the SMS texting system, as well as the preparatory activities that are necessary for its launch.

## Materials and methods

### System conceptualization and design

The system was designed on a district health information platform to received and interpreted the message sent from health facilities in predefined codes. Infrastructures included in the design included a server to host the database and store every text sent by each health facility, android based software designed to receive the text messages and forward to the server and an interpreter interface that links the messages from the HF to the DHIS2 platform. On this intervention, the health facilities are the clients, while the DHIS2 platform designed to present the data is the broker.

The system is built on an Android™ platform. At the end of each RI session, trained healthcare workers (HCWs) in each HF send SMS text to the phone number associated with the server. The phone number used to connect and transmit the SMS text identifies each HF. This system uses three servers for data storage: the SMS harvester, the interpreter, and the SMS dashboards server. The SMS harvester collects all SMS texts sent to the system and secures the data from loss. Once the text has been collected and saved, the information is pushed to the interpreter server, which translates the codes into a readable format. The SMS dashboard server populates the dynamic dashboard based on the programmed indicators.

Each HF summarizes the number of children vaccinated for each antigen and the number of vaccine doses in vials opened ("vaccine doses opened") during the session. This information is entered in SMS text using pre-determined codes and sent to a pre-assigned number which has been registered with service communication service providers in Nigeria for the purpose of this intervention. This intervention was designed with minimum financial impact on the health facilities such that each text message sent from a HF costs four Naira ($0.01) which is cost of SMS in Nigeria. However the financial burden of the intervention including the human resource, infrastructure and space was on the Government of Nigeria and partners supporting the intervention. The message is converted to data points and transmitted to a DHIS2 server at to automatically update a database/dashboard (see Fig 1). National, state and LGA level officers monitor the reporting system and give feedback to all HFs through phone calls and in-person supervisory visits.

### Dashboard design and utilization

The daily SMS reporting system was developed to enhance data reporting from HFs. Compared to the monthly reporting structure of the national health information system, District Health Information Software 2 (DHIS2), routine immunization module, the SMS platform encourages data reporting at the end of each RI vaccination session (fixed-post or outreach) The system was designed to automatically populate a dashboard built on DHIS2 platform that present routine immunization indicators with charts and figures monitoring the performance

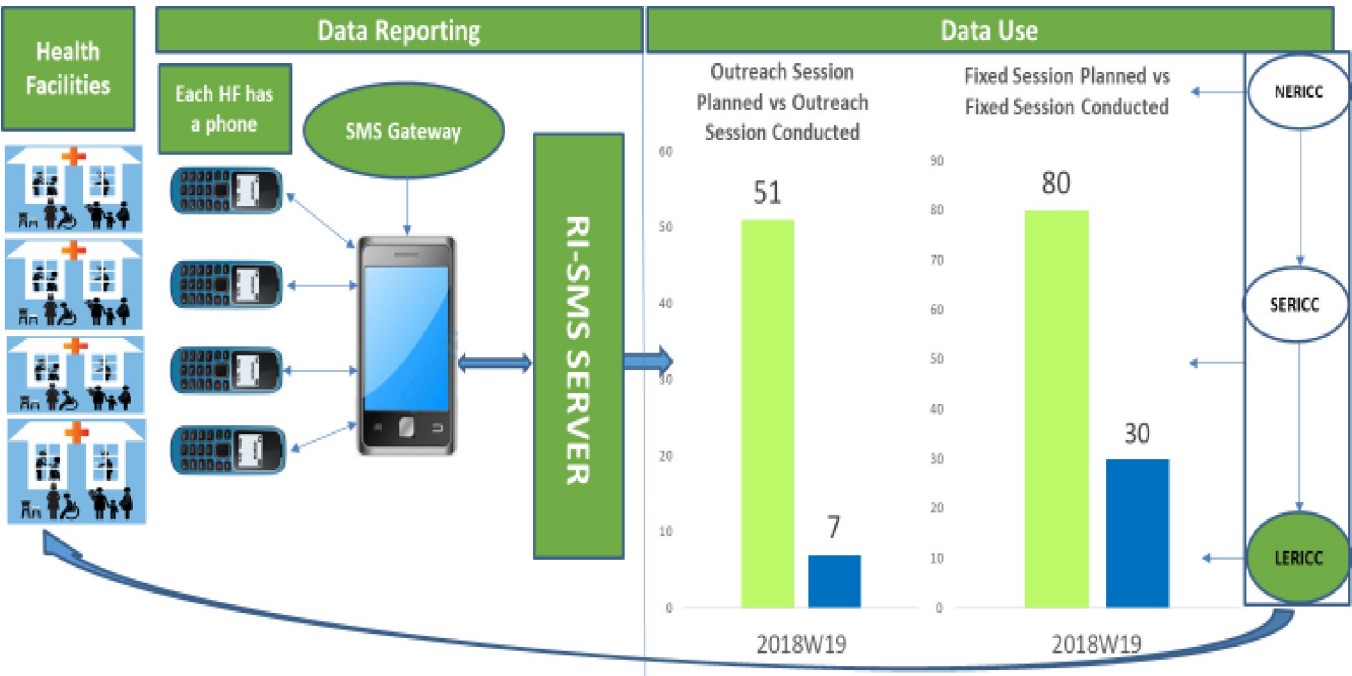

**Fig 1. Information flow chart for the SMS reporting system.**

of the programme district level, which may be comparable to those on the national DHIS2 RI dashboard. Some of the metrics on the dashboard included number of routine immunization session conducted out of the planned, number of children immunized by each vaccine schedule for instance Penta 1, Penta 2, Penta 3 against the target for each health facility has contained in the session plan, vaccine wastage. Based on the dashboard, the programme is monitored daily at the national, state and district levels to ensure that emerging issues are identified and addressed immediately.

## Content of the SMS text

Every HF offering RI services in the pilot was expected to transmit daily RI data using the SMS texting mechanism. Three sets of information were transmitted:

- Number of planned sessions per week (WK)

- Number of fixed-post sessions conducted (FS)

- Number of outreach session conducted (OS)

Two pieces of information were expected on the WK message, while 15 pieces of information were expected in each text for the FS and OS messages. The letter identifying each piece of information ("A" through "l") was associated with the number that answered the question on children immunized or vaccine doses opened. For example, if 15 children received a BCG dose and 30 received a measles dose, the codes are "a15" and "f30". Fig 2 reports a sample text message.

## Piloting

We piloted the SMS reporting strategy in two LGAs of Nasarawa state: one urban (Keffi, with 25 HFs) and one rural (Keana, with 30 HFs). The pilot lasted 151 days, starting in December

| code | Data Element Report | data |
|------|---------------------|------|
| a | Number of children given BCG | 15 |
| b | Number of children given Penta 1 | 6 |
| c | Number of children given Penta 2 | 3 |
| d | Number of children given Penta 3 | 6 |
| e | Number of children given IPV | 30 |
| f | Number of children given Measles | 10 |
| g | BCG doses opened for session | 20 |
| h | Penta doses opened for session | 30 |
| i | IPV doses opened for session | 10 |
| j | Measles opened for the session | 10 |
| k | Did you receive supportive supervision visit during this session from the LGA team | Yes = 1, No = 0 |
| l | Did you have any vaccine stock outs during the session | Yes = 1, No = 0 |

fs a15 b6 c3
d6 e30 f30 g10 h10 i30 j10
k1 l1

**Fig 2. Codes and sample message.**

2017 and ending in April 2018. A one-day training was conducted in November 2017 for 145 HCWs, including eight state officers, 11 LGA officers, 20 ward focal persons, and 106 HF staff. Many additional government personnel were trained to alleviate the issue of staff turn-over and better support the data collection and RI performance processes at the state and LGA levels. All HFs received a mobile phone with an activated SIM card to transmit data after every RI session. Once the data were entered, the dashboard was visible to all users with access privileges and reported all data points collected. Security issues limited the planned implementation of RI sessions in Keana LGA. Therefore, reporting was affected at different points during the pilot.

## Supervision

National-, state- and LGA-level officers monitored the output of the SMS reporting system and provided feedback to both LGA and HF officers. They offered daily and weekly feedback on gaps and weaknesses through phone calls and supervisory visits to the HFs. Two strategies were used for supervision during the pilot:

A. Review meetings: Conducted by a team from the national and state level, for HCWs and their supervisors. Supervisors compared the RI data reported through SMS with the paper-based immunization records, as well as the monthly reports from the DHIS2 platform for the same HF. An on-the-spot refresher training was conducted for the HF staff on the use of the codes and ensuring that messages were properly sent.

B. Field visits: The LGA-level teams visited the HFs each month, while the state and national team conducted intermittent visits. These visits were an opportunity to provide refresher training at the HF-level while conducting record verification based on submitted reports.

## Data analysis

Analyses were conducted using Microsoft Excel and RStudio 3.5.1. We reviewed the number of reports sent per day, so we could assess the weekly transmission trends and identify any recurring patterns. Also, we calculated the completeness rate for each HF by assessing the

number of daily reports transmitted over the 151 days of the pilot. Finally, we considered the quality of the RI data transmitted. Using these daily data points, we compared the reported number of vaccinated children with the reported number of doses administered, by antigen and by type of vaccination session (fixed-post vs. outreach). If the two numbers diverged, we assessed whether the difference was due to high wastage rate (doses administered > children vaccinated) or data entry errors (children vaccinated > doses administrated). Descriptive analysis was conducted using bar and line graphs to demonstrates trends and patterns in the data.

### Ethical approval

Approval for the pilot was received from the National Primary Health Care Development Agency (NPHCDA) and the Nasarawa State Primary Health Care Development Agency (SPHCDA). No consent was required from individual health facilities.

## Result

### Completeness rates

Of the 55 HFs selected, 50 (91%) were included in the pilot: 26 in Keana and 24 in Keffi. During the pilot, 47 HFs transmitted RI data via SMS, for an overall completeness rate of 85%. The pattern of reporting varied among types of HF, with hospitals (secondary care) (6%) reporting 5 times per week, urban primary health care centers (12%) reporting at least twice per week, and primary care HFs (82%) reporting at least once per week. These differences reflect the weekly vaccination schedule mandated for different types of facility (Fig 3).

Each week, the national-level server received an average of six SMS messages over the 33 expected for outreach RI sessions, and 30 SMS messages over the 66 expected for fixed-post RI sessions (see Fig 4). The expected values are derived from the vaccination schedule that each type of HF is assigned. After the first two weeks, reports were not submitted during weekends

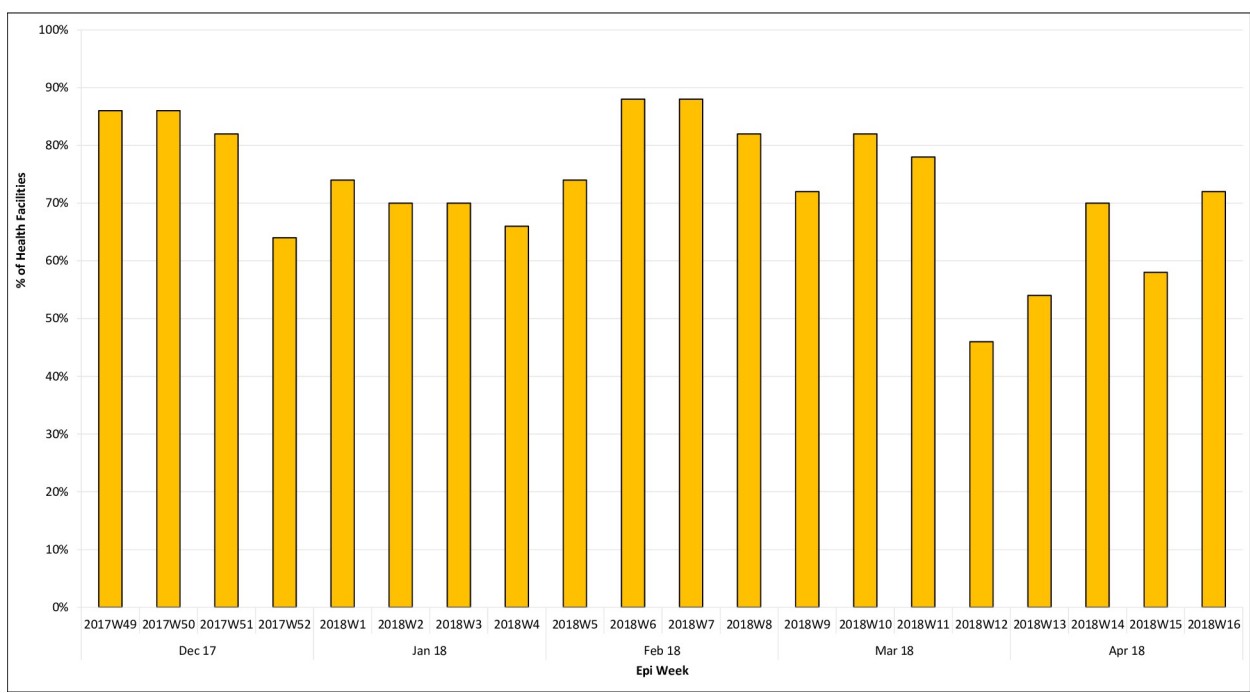

**Fig 3. Weekly reporting from HFs on the daily SMS platform, December 2017-April 2018.**

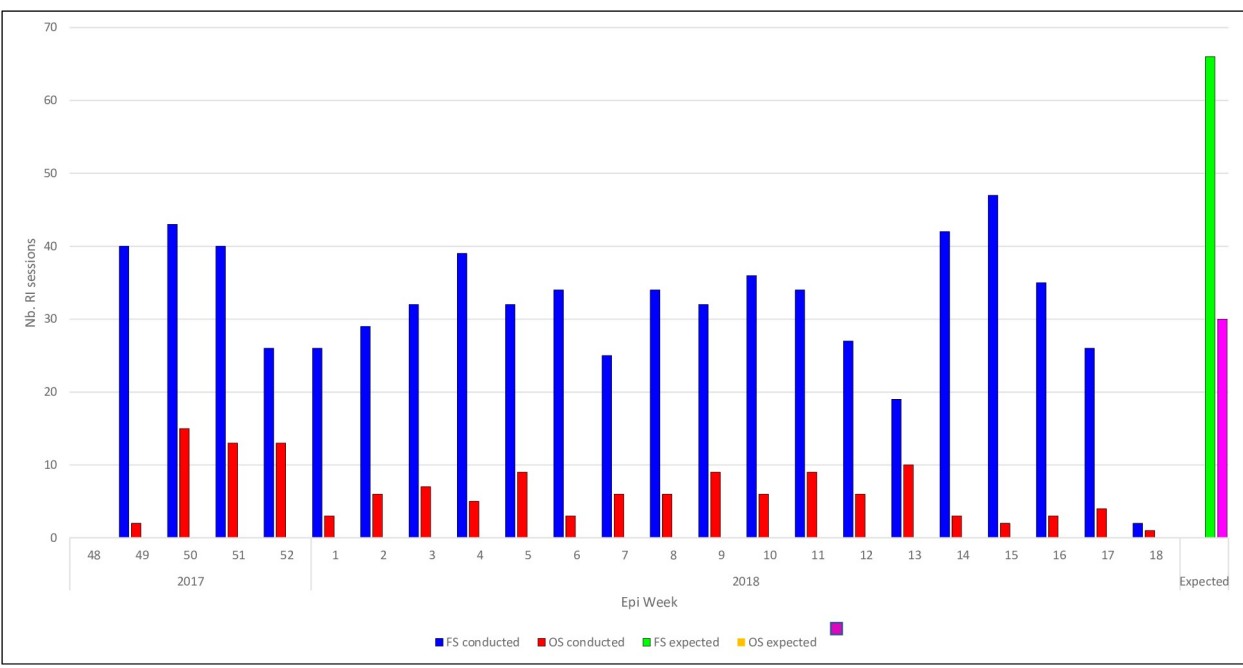

**Fig 4. Number of HFs that submitted RI reports through SMS texts each week, over a total of 66 possible entries per week for fixed-post sessions and 33 possible entries per week for outreach sessions (one for each HF included in the pilot).**

because no vaccination sessions were offered on those days. Also, we recorded the highest number of reporting HFs on Tuesdays, with fewer and fewer HF sending SMS texts in the following days. This pattern was consistent throughout the pilot (data not reported).

## Trends over time

We compare the number of doses available from the vials opened during one RI fixed-post session to the number of children reportedly vaccinated during that same fixed-post session. We consider the BCG and measles antigens (Figs 5 and 6, respectively). For both vaccines, the SMS reports indicate that the number of doses made available (or "opened") was always higher than the number of children vaccinated, as expected. For BCG, the reported number of opened doses and the number of vaccinated children was almost identical each week, suggesting minimal wastage of the antigen. For measles, instead, the wastage rate appears to be higher. For both vaccines, the two lines remain parallel despite the day-by-day fluctuations.

This report is aggregate across HFs, but allows us to assess whether there has been any change in the reports of either doses available or children vaccinated over time. There was no evidence of a trend in change in either.

## Supportive supervision

Fig 7 reports the number of fixed-post RI sessions that were reportedly conducted, as well as the number of those session that were supervised. The average number of fixed-post sessions conducted per day per HF was approximately 1, while the average number of sessions supervised each day is 0.3. This result suggests that, on average, at least one vaccination session in every three conducted is supervised by an LGA-level RI officer. Activities conducted during the visit include: RI data validation between submitted reports and paper registers, on-the-job

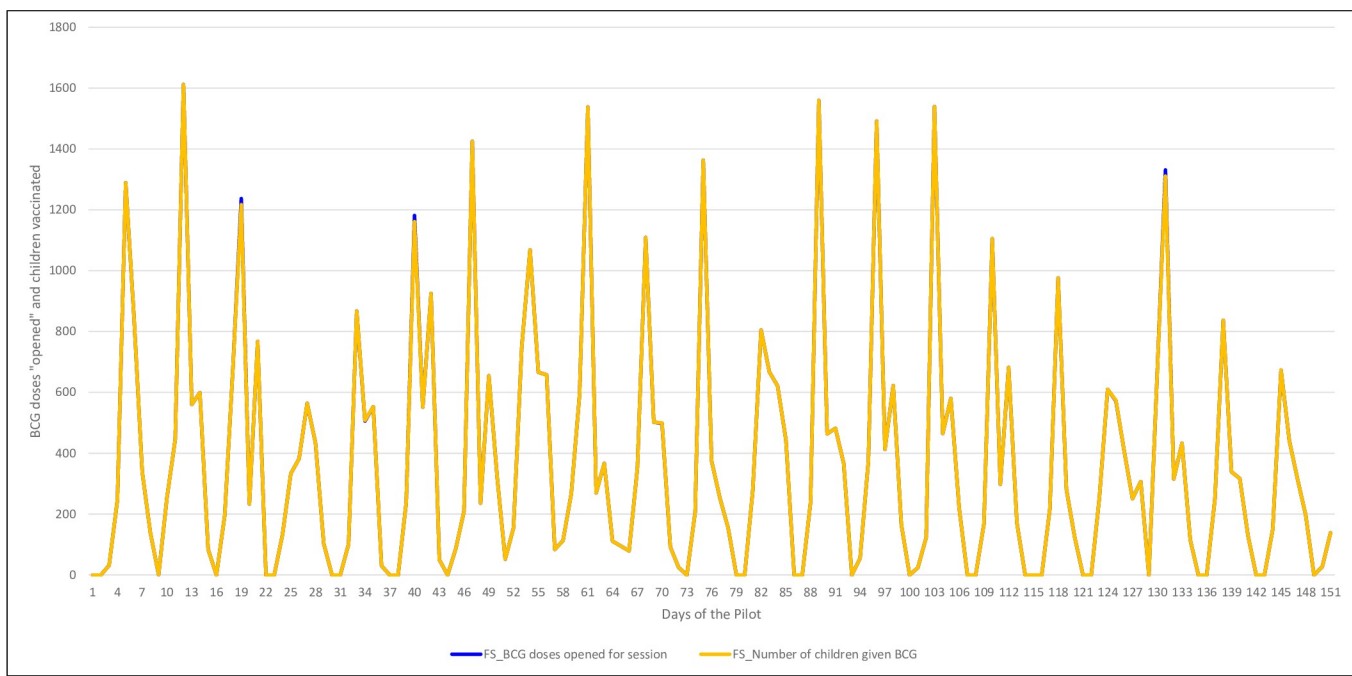

**Fig 5. Number of children vaccinated with BCG vs. number of BCG doses opened during fixed-post vaccination sessions, during the 151 days of the pilot.**

refresher trainings on reporting through the SMS platform, resolving any network and phone problems, and advocating to relevant authorities on resolving issues identified during visits.

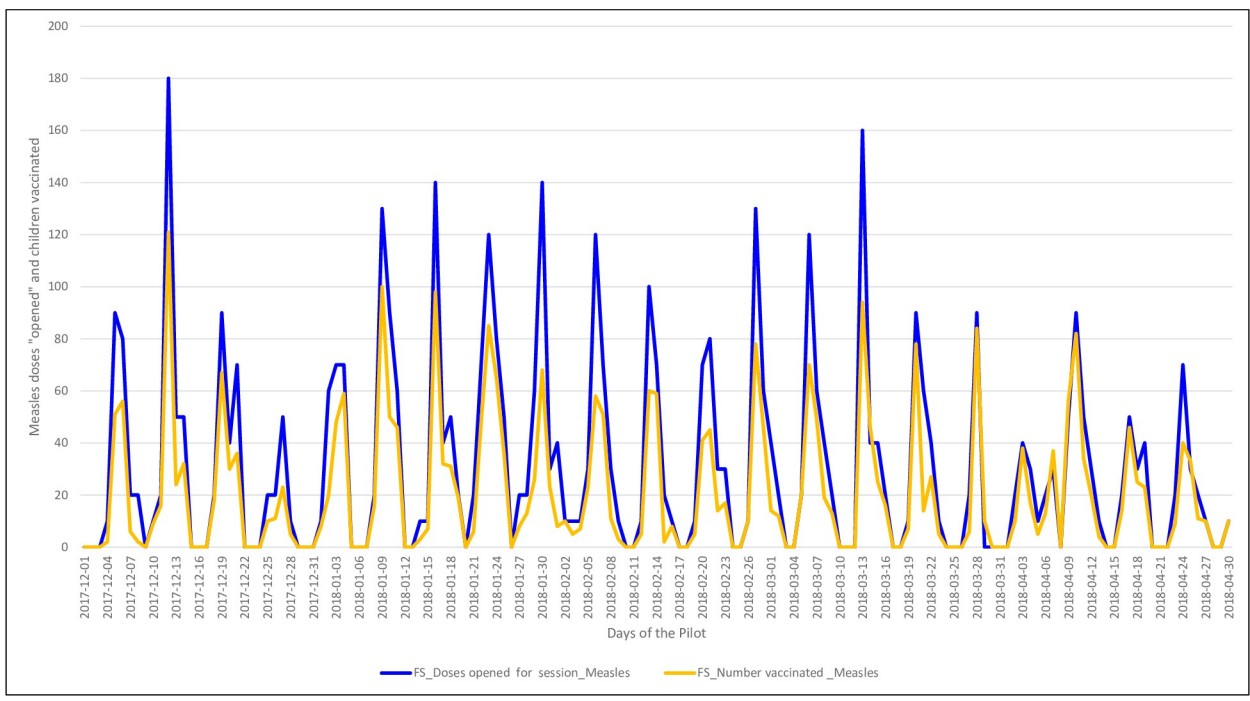

**Fig 6. Number of children vaccinated with measles vaccine vs. number of measles vaccine doses opened during fixed-post vaccination sessions, during the 151 days of the pilot.**

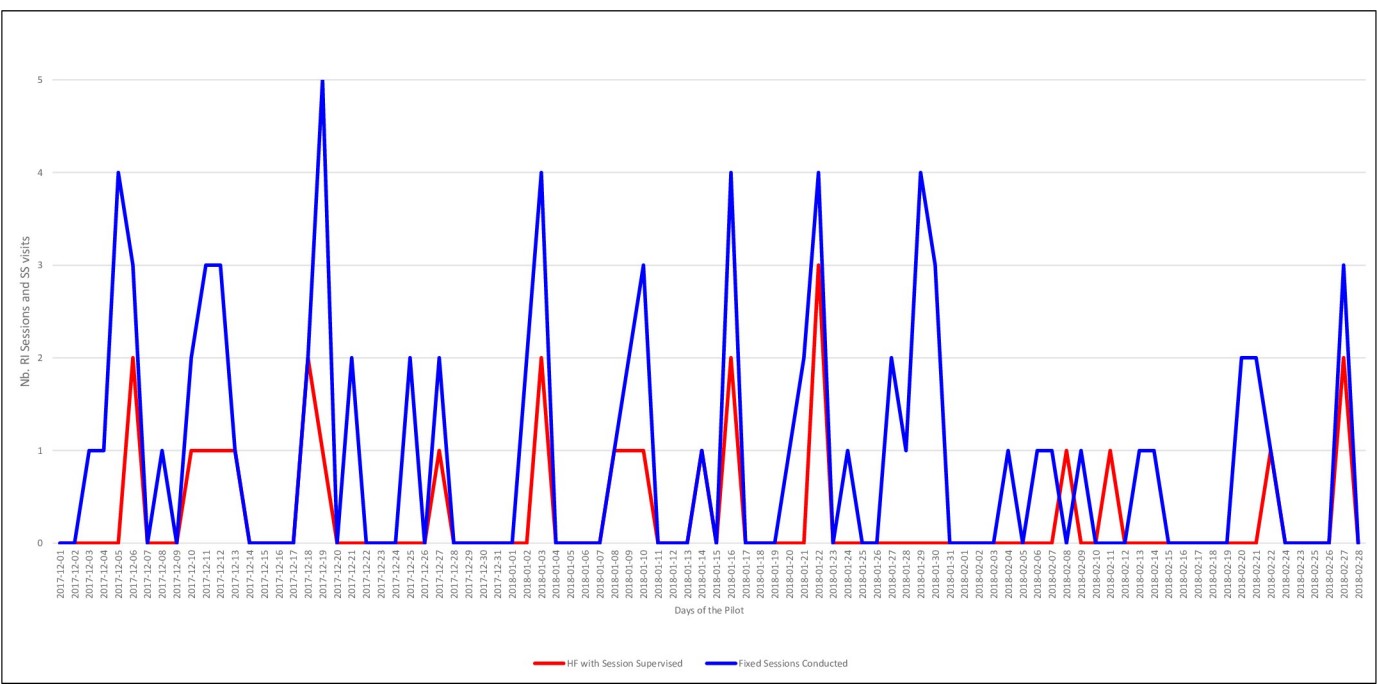

**Fig 7. Number of fixed-post RI sessions reportedly conducted vs. number of those sessions that received supportive supervision, per day.**

## Assessment for data consistency

For each HF, we compared the number of vaccinated children to the number of antigen doses reportedly opened over the course of the pilot, using measles vaccine as an example. Our goal was to assess whether these number coincide, since each child should receive a single dose of measles vaccine and some wastage of doses opened is expected. These comparisons helped us assess the internal consistency of the data, given that data are supposedly collected and transmitted in real-time each day. The number of doses opened during fixed-post sessions (see Fig 8A) and outreach sessions (see Fig 8B) were analyzed separately.

During the fixed-post sessions of 24 (51%) of the 47 HFs with RI reports, we noticed that the number of vaccinated children was reported to be higher than the number of measles vaccine doses contained in open vials during fixed-post sessions. During the outreach sessions organized by two HF, there were more children reportedly vaccinated than measles vaccine doses administered. Also, 17 HFs reported administering no measles vaccine doses during outreach sessions. There are two instances where the reported wastage rate is above 80%.

## Discussion

The pilot demonstrated that HFs can use SMS texts to report the results of daily RI sessions. Compared with DHIS2, for which there are only end-of-the-month reports, daily SMS reporting can be used to monitor RI data more frequently and facilitate timely interventions for improving the quality of data collected by the program. These data can also provide supervisors with another source of information to identify areas where RI services are suboptimal. In our pilot, decision-makers were able to identify which HFs did not report sessions conducted and ensure prompt follow-up. Each week, the national-level server received an average of six SMS messages of the 33 expected for outreach RI sessions, and 30 SMS messages of the 66 expected for fixed-post RI sessions. Several studies have explored the use of SMS texting for

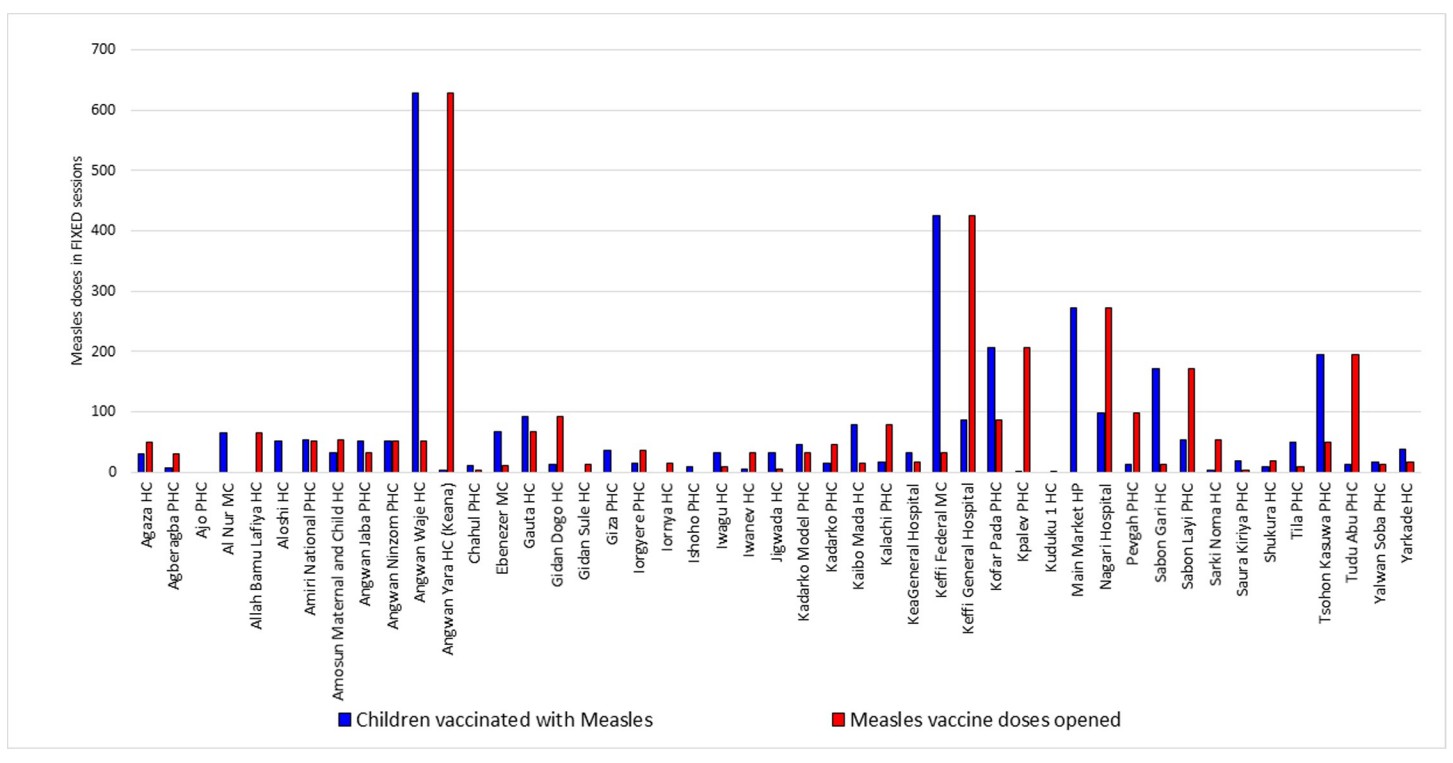

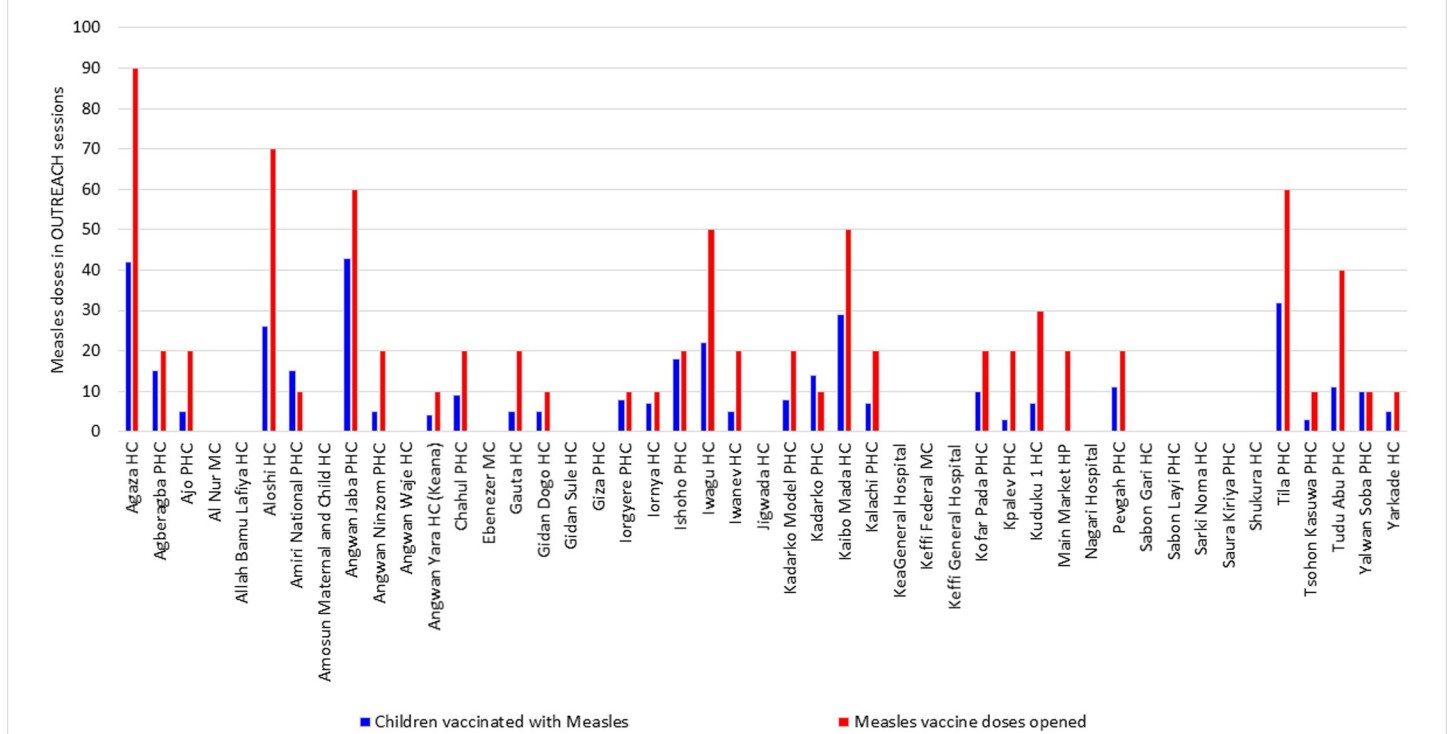

**Fig 8. a.** Number of children vaccinated with Measles vs. number of measles vaccine doses opened during the pilot, as part of <u>fixed-post</u> vaccination sessions. **b.** Number of children vaccinated with Measles vs. number of measles vaccine doses opened during the pilot, as part of <u>outreach</u> vaccination sessions.

improving RI data quality and uptake of immunization services [11–14], however, none focused on using the data to monitor program performance.

We identified multiple challenges. First, our SMS texting pilot was planned to launch in 55 HFs of two LGAs in Nasarawa state however among the HFs selected, insecurity in eight HFs in Keana LGA precluded conducting any vaccination sessions, thus participating in the pilot. Also, there were no significant improvements in completeness rate over time. We also assessed that the reports on the number of children vaccinated often did not coincide with the reported number of doses opened. In fact, often the number of children vaccinated was higher than the number of doses used. This situation was found for both antigens considered (BCG and measles) and across both fixed-post and outreach vaccination sessions. These results suggest that both HF program implementation and the reporting system still requires considerable support–both technical and logistical–before there can be consistent and reliable RI data points.

When we consider time trends, the reported relationship between doses administered and children vaccinated remains consistent across the 151 days of the pilot for both the BCG and measles antigens. There were daily fluctuations in the number of HFs that submitted a report, with peaks on Tuesdays. The wastage rate remained consistent day by day.

With regards to the accuracy of the RI data submitted through SMS texts, the main concern was the discrepancy we noted between the reported number of vaccines doses opened and the reported number of children immunized. Specifically, a few HFs reported wastage rates above 80% for BCG (15 HFs) and measles (two HFs). More concerning is that 18 HFs (36%) reported a higher number of children vaccinated than doses available from the vials opened for both BCG and measles. Since this is obviously impossible, we must consider whether these data entry mistakes are the result of human error or intentional attempts at data falsification. Unfortunately, the national-level staff who conducted the six supportive supervisory visits during the pilot did not examine this matter, and therefore we cannot assess whether these errors were voluntary or not. Nonetheless, this pilot helped supervisors identify this major issue in RI reporting.

This pilot highlighted some limitations of this SMS texting approach. Observed fluctuations in daily SMS completeness were mostly due to challenges with the HFs' unfamiliarity with the SMS platform and its transmission requirements. No job aids were available at the HFs to guide the compilation of the data and the composition of the SMS text. Weak telecommunication networks in certain areas of LGAs contributed to suboptimal completeness rates as well, and server down-time contributed to delayed transmission. Implantation teams needed more time than planned to conduct all systems monitoring procedures. To address these issues, the implementation teams printed job aids that were distributed by national-level supervisors to all HFs in the pilot. The team managing the database at the national level received feedback how to deal with system down-time. Also, the national team identified the telecommunication network with the best connectivity in the area, and advised the HFs to use that provider for reporting. These activities resulted in improved monitoring of RI sessions held over time and children reached with vaccination. Lastly among limitations, the cost of data transmission was an important barrier to reporting completeness. Despite the low cost of forwarding a text, the HFs were not given phone credit to use during this pilot.

Despite these challenges and limitations, this pilot demonstrated that HFs can use SMS texts to report the results of daily RI sessions that can be useful for prompt actions. If analyses are stratified both by day of report and by HF, SMS texting can help identify inaccurate reporting for targeted and prompt on-the-job refresher trainings. Also, tracking vaccine doses available for planned sessions could help prevent stock-outs and vaccine shortages. For long-term sustainability and independence from of external support, states and LGAs will need to take ownership of the SMS reporting platform–including payment of phone credit options and maintenance of the mobile fleet. Plans are under development to resolve these challenges, so the completeness and accuracy of reporting can improve. Because of the experience of this SMS pilot project in Nasarawa, the Nigeria NPHCDA has now scaled this process to all LGAs in 18 states.

## Conclusion

Real time RI data transmission by SMS test is feasible in Nigeria, and HCWs can implement this transmission process after thorough training. Encouraging HCWs to use the daily SMS reporting system could improve the completeness of RI data reporting and consequently facilitate prompt decision-making at different levels. Decision makers can identify which HFs have not reported their RI data and intervene promptly to identify the reason and which HFs are not meeting performance criteria. This system enhances the prompt identification of gaps in RI service delivery, informs the strategies needed to address these gaps, complements the quality of RI data in Nigeria, and ultimately has the potential to increase RI coverage across the country. Nigeria has now scaled this process to 18 states and all their LGAs, and results of this expansion are pending.

## Supporting information

**S1 Dataset.**
(ZIP)

## Acknowledgments

We wish to acknowledge everyone who contributed to the implementation of the Nasarawa pilot, including members of National Emergency Routine Immunization Coordinating Center (NERICC), members of the Nasarawa State Emergency Routine Immunization Coordinating Center (SERICC) and the LGA teams who ensured the proper application of the guidelines.

## Author Contributions

**Conceptualization:** Adekunle Akerele, Belinda Uba, Sulaiman Etamesor, Oluwasegun Joel Adegoke, Okposen Bassey, Faisal Shuaib.

**Data curation:** Adekunle Akerele, Ameh Enyojo, Friday Josiah.

**Formal analysis:** Adekunle Akerele, Margherita E. Ghiselli.

**Funding acquisition:** Margherita E. Ghiselli, Omotayo Bolu.

**Investigation:** Belinda Uba, Jamila A. Umar, Olorunsogo Bidemi Adeoye, Esther Ayandipo, Okposen Bassey, Waziri Ndadilnasiya.

**Methodology:** Adekunle Akerele, Belinda Uba, Matthew Aduloju, Sulaiman Etamesor, Jamila A. Umar, Olorunsogo Bidemi Adeoye, Friday Josiah, Esther Ayandipo, Itse Olaoye, Oluwasegun Joel Adegoke, Sampson Sidney, Okposen Bassey, Waziri Ndadilnasiya.

**Project administration:** Adekunle Akerele, Matthew Aduloju, Sulaiman Etamesor, Itse Olaoye, Sampson Sidney, Murtala Bagana, Okposen Bassey, Waziri Ndadilnasiya, Omotayo Bolu.

**Resources:** Murtala Bagana, Omotayo Bolu.

**Software:** Adekunle Akerele, Matthew Aduloju, Olorunsogo Bidemi Adeoye, Oluwasegun Joel Adegoke.

**Supervision:** Adekunle Akerele, Belinda Uba, Sulaiman Etamesor, Jamila A. Umar, Ameh Enyojo, Friday Josiah, Itse Olaoye, Oluwasegun Joel Adegoke.

**Validation:** Adekunle Akerele, Jamila A. Umar, Ameh Enyojo, Esther Ayandipo, Margherita E. Ghiselli.

**Visualization:** Matthew Aduloju, Jamila A. Umar, Olorunsogo Bidemi Adeoye, Itse Olaoye.

**Writing – original draft:** Adekunle Akerele, Belinda Uba, Sulaiman Etamesor.

**Writing – review & editing:** Adekunle Akerele, Belinda Uba, Esther Ayandipo, Itse Olaoye, Oluwasegun Joel Adegoke, Sampson Sidney, Murtala Bagana, Okposen Bassey, Margherita E. Ghiselli, Waziri Ndadilnasiya, Omotayo Bolu, Faisal Shuaib.

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
