## [Decision Letter · Decision Letter 0]

6 Mar 2021

PONE-D-20-23584

Improving Routine Immunization Data Quality using Daily Short Message System Reporting Platform: An Experience from Nasarawa State, Nigeria

PLOS ONE

Dear Dr. Akerele,

Thank you for submitting your manuscript to PLOS ONE. After careful consideration, we feel that it has merit but does not fully meet PLOS ONE’s publication criteria as it currently stands. Therefore, we invite you to submit a revised version of the manuscript that addresses the points raised during the review process.

In particular, Reviewer 1 has raised some concerns that need careful consideration. 

We look forward to receiving your revised manuscript.

Kind regards,

Michele Tizzoni

Academic Editor

PLOS ONE

Journal Requirements:

"None"

Reviewers' comments:

Reviewer's Responses to Questions

**Comments to the Author**

1. Is the manuscript technically sound, and do the data support the conclusions?

Reviewer #1: Partly

Reviewer #2: Yes

2. Has the statistical analysis been performed appropriately and rigorously? 

Reviewer #1: No

Reviewer #2: Yes

3. Have the authors made all data underlying the findings in their manuscript fully available?

Reviewer #1: No

Reviewer #2: Yes

4. Is the manuscript presented in an intelligible fashion and written in standard English?

Reviewer #1: Yes

Reviewer #2: Yes

5. Review Comments to the Author

Reviewer #1: The manuscript describes an assessment of data quality and cross checking data regarding to routine immunization in Nigeria through a pilot of SMS-based platform.

Although the paper is well-written (except for some typos, lower case, etc, present in the text), it presents a lack of several relevant aspects such as better overview in the initial proposed assumption. First, I would suggest a better characterization of the problem, bringing contributions of mobile health, specifically in the aspects of SMS-based communication systems. There is a huge variety of papers describing the role and impact of SMS interventions related to health. Moreover, SMS really constitutes a large spectrum of applications in health interventions, from reminders or daily checking to nudges and behavioral change. That is why would be really helpful having a better characterization of the essence of this study with other similar benchmarks. An example that reinforce this idea is the unproper way the reference 12 was placed. In that case, the object which the reference worked on it is very different of the current proposal, making difficult to support the assumption created in that part. I suggest to make a better review of it.

A SMS system requires a non-simple workflow involving brokers and vendors, which implies in such limitations in terms of allocated budget to carry out SMS interventions. It was not clear that SMS text cost informed ($0.01) is about the final price of vendor or the SMS landed in the system. This really matters if the paper will be used to disclose a metholodogy in order to be reproducible by other countries that have similar challenges. Also to understand the overall costs including the fixed-cost of an unique number for each HF it might be useful to bring more reality of financial impact of it.

In the section "System programming development" actually discloses almost nothing about the programming development per se. If the idea is to bring technical details such workflow, programming languages, packages, modules, routines, tasks, data wrangling and data visualization, it would be recommended to bring more details about it. On the other hand, if the authors want to avoid technical terms, then the title of section should be changed.

About the dashboard, I suggest include which were the set of metrics that were showed into it. It was only descriptive statistics or something more sophisticated was used? Any sort of early warning system was deployed? Any framework (D3.js, Shiny.r, Exploratory.io, PowerBI, Tableau, QlikView, etc) was used?

I think there is a lack of results in terms of metrics. I was expecting to see production metrics to prove how the system worked and scaled. For example, retention or engagement metrics are missing in the text and I'm pretty sure the authors have the raw data to calculate those.

For the analysis, why was not used any sort of simple smoothing like LOESS or GAM, or even rolling average, in order to reduce the fluctuation and bring a more consistent visualization of the series?

A network coverage map would be useful to understand if all of HF areas were in similar covered regions, bringing more equality in the intervention.

The assumptions "For long-term sustainability independent of external support, states and

LGAs will need to take ownership of the SMS reporting platform – including payment of phone credit options and maintenance of the mobile fleet." seems to simple considering the complexity of the issue. This sort of coordination sometimes is not only about funding or training, instead, if the decision makers are not involved understanding the actionable data that has been generated, this ownership will never come. I would suggest to reprhase this sentence in order to explore better the issues and show potential avenues for resolution of it.

This sentence "Our results can be useful for other countries’ programs that are considering SMS texting as a real-time information tool of RI activities." is not super accurate, since the presented results do not support 100% that. Also, to generate an evidence that supports this assumptions, would be necessary observe the counterfactuals as well, which was not present in this study.

Reviewer #2: In this work, the authors investigate a timely problem of data quality issues around routine immunization (RI) data monitoring. The paper describes the launch of the RI data collection system by SMS texting and analyzes the first data points collected.

The paper is interesting and clearly written. The SMS texting strategy has been used for the first time here to collect RI data. This paper represents an essential contribution to helping other countries that are considering implementing this kind of strategy. Besides, it highlights the significant future steps that are required to improve this type of system. Indeed, the results are promising, but some further work is needed to reach more optimal and reliable results.

I have made some minor comments for both the introduction and the methods section. Also, I found some typos in the manuscript and reported them in the “specific comments” section.

Overall, I think that this work is suitable for publication after addressing the minor comments.

6. PLOS authors have the option to publish the peer review history of their article (what does this mean?). If published, this will include your full peer review and any attached files.

Reviewer #1: **Yes: **Onicio B Leal-Neto

Reviewer #2: No

---

## [Author Response · Author response to Decision Letter 0]

6 May 2021

To editor, all comments has be addressed

Nigerian National Primary Health Care Development Agency supported the project with dedicated staff for the project management. No direct funding was received for the project.

---

## [Decision Letter · Decision Letter 1]

18 May 2021

PONE-D-20-23584R1

Improving Routine Immunization Data Quality using Daily Short Message System Reporting Platform: An Experience from Nasarawa State, Nigeria

PLOS ONE

Dear Dr. Akerele,

Thank you for submitting your manuscript to PLOS ONE. After careful consideration, we feel that it has merit but does not fully meet PLOS ONE’s publication criteria as it currently stands. Therefore, we invite you to submit a revised version of the manuscript that addresses the points raised during the review process.

Please, take into account the last minor comments made by Reviewer #1 in your revision.

We look forward to receiving your revised manuscript.

Kind regards,

Michele Tizzoni

Academic Editor

PLOS ONE

Journal Requirements:

Reviewers' comments:

Reviewer's Responses to Questions

**Comments to the Author**

1. If the authors have adequately addressed your comments raised in a previous round of review and you feel that this manuscript is now acceptable for publication, you may indicate that here to bypass the “Comments to the Author” section, enter your conflict of interest statement in the “Confidential to Editor” section, and submit your "Accept" recommendation.

Reviewer #1: All comments have been addressed

Reviewer #2: All comments have been addressed

2. Is the manuscript technically sound, and do the data support the conclusions?

Reviewer #1: Partly

Reviewer #2: Yes

3. Has the statistical analysis been performed appropriately and rigorously? 

Reviewer #1: No

Reviewer #2: Yes

4. Have the authors made all data underlying the findings in their manuscript fully available?

Reviewer #1: Yes

Reviewer #2: Yes

5. Is the manuscript presented in an intelligible fashion and written in standard English?

Reviewer #1: Yes

Reviewer #2: Yes

6. Review Comments to the Author

Reviewer #1: The authors have addressed the majority of comments I've made in the first review, however some minor questions are still opened as study's limitation or scope. For having a more consistent text, I would recommend try to addressing that. They are the same from the first review.

Reviewer #2: Minor edit in discussion page 8:

1. "For long-term sustainability and independence from of external support" : "of" seems redundant

7. PLOS authors have the option to publish the peer review history of their article (what does this mean?). If published, this will include your full peer review and any attached files.

Reviewer #1: **Yes: **Onicio Batista Leal Neto

Reviewer #2: No

---

## [Author Response · Author response to Decision Letter 1]

2 Jul 2021

We responded to all minor revisions of the reviewers

---

## [Decision Letter · Decision Letter 2]

21 Jul 2021

Improving Routine Immunization Data Quality using Daily Short Message System Reporting Platform: An Experience from Nasarawa State, Nigeria

PONE-D-20-23584R2

Dear Dr. Akerele,

We’re pleased to inform you that your manuscript has been judged scientifically suitable for publication and will be formally accepted for publication once it meets all outstanding technical requirements.

Kind regards,

Michele Tizzoni

Academic Editor

PLOS ONE

Additional Editor Comments (optional):

Reviewers' comments:

Reviewer's Responses to Questions

**Comments to the Author**

1. If the authors have adequately addressed your comments raised in a previous round of review and you feel that this manuscript is now acceptable for publication, you may indicate that here to bypass the “Comments to the Author” section, enter your conflict of interest statement in the “Confidential to Editor” section, and submit your "Accept" recommendation.

Reviewer #1: All comments have been addressed

Reviewer #2: All comments have been addressed

2. Is the manuscript technically sound, and do the data support the conclusions?

Reviewer #1: Yes

Reviewer #2: Yes

3. Has the statistical analysis been performed appropriately and rigorously? 

Reviewer #1: No

Reviewer #2: Yes

4. Have the authors made all data underlying the findings in their manuscript fully available?

Reviewer #1: No

Reviewer #2: Yes

5. Is the manuscript presented in an intelligible fashion and written in standard English?

Reviewer #1: Yes

Reviewer #2: Yes

6. Review Comments to the Author

Reviewer #1: (No Response)

Reviewer #2: I approved the manuscript after revision number one. In my opinion, the manuscript is now ready for publication

7. PLOS authors have the option to publish the peer review history of their article (what does this mean?). If published, this will include your full peer review and any attached files.

Reviewer #1: **Yes: **Onicio Batista Leal Neto

Reviewer #2: No

---

## [Editor Report · Acceptance letter]

28 Jul 2021

PONE-D-20-23584R2 

Improving Routine Immunization Data Quality using Daily Short Message System Reporting Platform: An Experience from Nasarawa State, Nigeria 

Dear Dr. Akerele:

I'm pleased to inform you that your manuscript has been deemed suitable for publication in PLOS ONE. Congratulations! Your manuscript is now with our production department. 

Kind regards, 

on behalf of

Dr. Michele Tizzoni 

Academic Editor

PLOS ONE